# UDH: Universal Deep Hiding for Steganography, Watermarking, and Light Field Messaging

**Chaoning Zhang**[*]
KAIST
chaoningzhang1990@gmail.com

**Philipp Benz**[*]
KAIST
pbenz@kaist.ac.kr

**Adil Karjauv**[*]
KAIST
mikolez@gmail.com

**Geng Sun**
KAIST
tosungeng@gmail.com

**In So Kweon**
KAIST
iskweon77@kaist.ac.kr

## Abstract

Neural networks have been shown effective in deep steganography for hiding a full image in another. However, the reason for its success remains not fully clear. Under the existing cover ($C$) dependent deep hiding (DDH) pipeline, it is challenging to analyze how the secret ($S$) image is encoded since the encoded message cannot be analyzed independently. We propose a novel universal deep hiding (UDH) meta-architecture to disentangle the encoding of $S$ from $C$. We perform extensive analysis and demonstrate that the success of deep steganography can be attributed to a frequency discrepancy between $C$ and the encoded secret image. Despite $S$ being hidden in a cover-agnostic manner, strikingly, UDH achieves a performance comparable to the existing DDH. Beyond hiding one image, we push the limits of deep steganography. Exploiting its property of being *universal*, we propose universal watermarking as a timely solution to address the concern of the exponentially increasing number of images and videos. UDH is robust to a pixel intensity shift on the container image, which makes it suitable for challenging application of light field messaging (LFM). Our work is the first to demonstrate the success of (DNN-based) hiding a full image for watermarking and LFM. Code: https://github.com/ChaoningZhang/Universal-Deep-Hiding

## 1 Introduction

The craft of steganography describes the secret communication without revealing the transported information to a third-party [25, 27, 14, 28]. The challenge for image steganography is to hide more information while keeping the container image look natural [17, 10, 9]. Recently, deep neural networks [32] have been shown to successfully hide a full image in another one [2] with a message capacity of 24 bits per pixel (bpp) significantly exceeding that of traditional techniques, *e.g.* HUGO [39] hides < 0.5 bpp. The task of (image) "steganography" with traditional techniques often requires perfectly decoding the secret message while remaining undetected by steganalysis [40]. In contrast, deep steganography in [2] has introduced a conceptually similar but technically different task of hiding a full image. Specifically, it relaxed the constraint of perfect decoding while focused on a high hiding capacity with a visual quality trade-off between container image and decoded secret image [2]. Due to the large hiding capacity, it is unlikely that the hidden image can remain undetected [3]. This new task has also been explored in a wide range of works [45, 47]. Acknowledging the difference between traditional steganography and deep steganography, in this work we adopt the term "deep

---

[*]Equal contribution

steganography" to be consistent with [2, 45, 47, 46]. The success of deep steganography also inspired the exploration of hiding binary information in deep watermarking [55] and deep photographic steganography, also termed light filed messaging (LFM) [46]. Despite large information capacity, deep steganography has a high visual quality, the reason of which remains yet unexplored. With the focus of hiding a secret image, our work is the first one towards explaining how deep steganography works as well as investigating it for applications in watermarking and LFM.

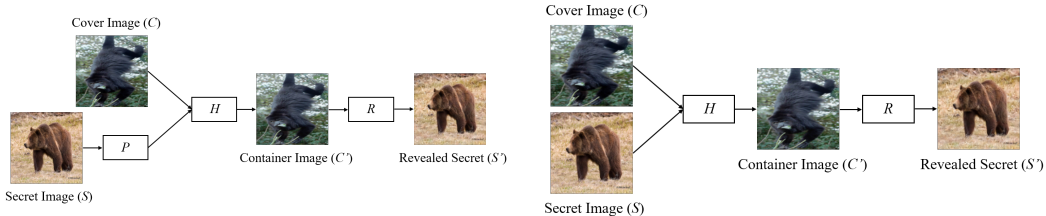

Figure 1: Existing DDH meta-architecture with (left) [2] or without(right) [45] $P$ network.

In this work, the general practice of hiding one image in another one is termed *deep hiding* which serves as a hypernym or umbrella term including deep steganography, watermarking and LFM. The existing deep hiding pipelines fall into one meta-architecture category termed cover-dependent deep hiding (DDH). As shown in Figure 1, the cover image ($C$) and (processed) secret image ($S$) are concatenated as the input of a *hiding (H)* network to generate a container image ($C'$). Another *reveal (R)* network is used to recover the secret image ($S'$). The objective is to minimize $||C' - C||$ and $||S' - S||$ simultaneously. Given $C'$ remains natural-looking, *i.e.* $||C' - C||$ is so small that it is human imperceptible, it is striking that the *reveal (R)* network can decode $S'$ almost perfectly from $C'$ [2]. The phenomenon of imperceptible hidden information triggering the $R$ network echos with a parallel research line of *adversarial attack* [42, 18, 48, 21, 5, 8, 7, 1, 15], where a small imperceptible perturbation fools a target network. More intriguingly, a single image-agnostic perturbation is found to exist for attacking most images and often called *universal adversarial perturbations* [35, 36, 23, 49, 50, 6]. Inspired by this, we explore the possibility to hide an image in a cover-agnostic manner, *i.e.* universal deep hiding (UDH).

The primary motivation of UDH is to facilitate explaining the success of deep steganography [2]. One natural guess is that messages are hidden in the least significant bits (LSB) [10], however, preliminary analysis in [2] rules out this possibility. Intuitively, $S_e = C' - C$ represents how $S$ is encoded in $C'$, however, it is not meaningful to analyze $S_e$ independent of $C$ in the existing DDH because $S_e$, (being equal to $H(C, S) - C$), is dependent on $C$. Since $S$ is encoded in $C'$, one alternative is to analyze $C'$ as a whole but the magnitude dominance of $C$ over $S_e$ makes it impractical. The above reasons complicate the exploration of how $S$ is encoded under the existing DDH. In the proposed UDH (See Figure 2), $S_e$ (being equal to $H(S)$) is independent of $C$. Thus, $S_e$ can be analyzed directly, which is a noticeable merit of UDH for understanding where and/or how the $S$ is encoded. We find that the success of UDH can be directly attributed to a frequency discrepancy between $S_e$ and $C$. With a cross-test of $H$ and $R$ from DDH and UDH, we also successfully demonstrate how DDH works.

Overall, compared with DDH, UDH is a more challenging task because the algorithm of UDH can not adaptively encode $S_e$ based on $C$. Empirically, however, we find that UDH results in a more smooth training and achieves comparable performance for deep steganography. Beyond hiding one image, we further push the limits of deep steganography with higher hiding capacity. Exploiting its property of being universal for high efficiency, we are the first to investigate and demonstrate the possibility of (DNN-based) *universal* watermarking. This can be a timely solution for efficient watermarking tackling the exponentially increasing number of images or videos. In contrast to HiDDeN [55] which watermarks by hiding binary information, we are the first to demonstrate (DNN-based) watermarking by hiding images. The UDH for hiding images without retraining can be readily extended to hide simple binary information, achieving superior performance than [55]. UDH is robust to pixel intensity shift on $C'$, which makes it more suitable for the task of LFM. In contrast to [46] that only hides binary information, UDH is the first to successfully hide and transmit an image robust to light effect, increasing its real-world applicability. It is also worth mentioning that UDH does not require collecting a large screen-pair dataset (1.9TB) as in [46]. For transmitting simple binary information, UDH achieves significantly better performance than [46].

## 2  Related work

Traditional steganography and watermarking have been extensively studied in [44, 34, 12, 16, 41, 33, 20], and we refer the readers to [4, 11] for an overall review. Our work focuses on understanding and harnessing deep learning for hiding messages in images and we summarize its recent advancement.

**Hiding a binary message in an image.** With their great success in a wide range of applications [53, 29, 30, 38, 52], DNNs also found adoption in steganography and watermarking [22]. In early explorations, DNNs have been adopted to mainly substitute a single stage of a larger pipeline [24, 26, 37]. Recently, the trend is to train networks end-to-end for the whole working pipeline. Hayes *et al*. first trained DNNs with adversarial training to hide binary messages in an end-to-end manner [19]. Taking robustness into account, HiDDeN [55] explored hiding binary messages for watermarking. Adversarial training was adopted in HiDDeN to minimize the artificial effect on $C'$. By encoding hyperlinks into binary bits, a concurrent work [43] also shows that DNNs can be trained to perform a robust encoding and decoding for physical photographs. The performance of these approaches can be evaluated by various metrics, such as capacity, secrecy, and robustness. There is often an inherent conflict between these metrics [19, 55]. For example, models with high capacity have low secrecy since hiding more information results in larger distortions on images. The models that are robust to distortions tend to sacrifice both secrecy and capacity. To increase robustness for watermarking, the hiding capacity in HiDDeN was less than 0.002 bpp [55].

**Hiding an image message in an image.** Hiding binary messages with DNNs has a low information capacity (typically lower than $0.5$ bpp), which does not fully exploit the potential of deep hiding. In a seminal work [2], deep steganography has been shown to hide a full image with a very high capacity of $24$ bpp. It adopted an additional preparation ($P$) network to process the image into a new form before concatenating it with the cover image, see Figure 1 (left). The technique of hiding an image in another can be easily extended to hide videos in videos, by sequentially hiding each frame of one video in the frame of another video. This approach has been explored in [45] where temporal redundancy has been exploited to hide the residual secret frame instead of the original image frame. Hiding 8 frames in 8 frames has also been explored in [47] where 3D-CNN is used to exploit the motion relationship between frames. Despite architecture differences of $H$ and $R$, prior works [45, 47] can be seen as an extension of [2] by excluding the $P$ network, see Figure 1 (right). Different from prior arts, our work is based on the proposed UDH meta-architecture, focusing on explaining the deep steganography success and investigating (universal) watermarking and LFM by hiding a secret image.

## 3  Universal deep hiding meta-architecture

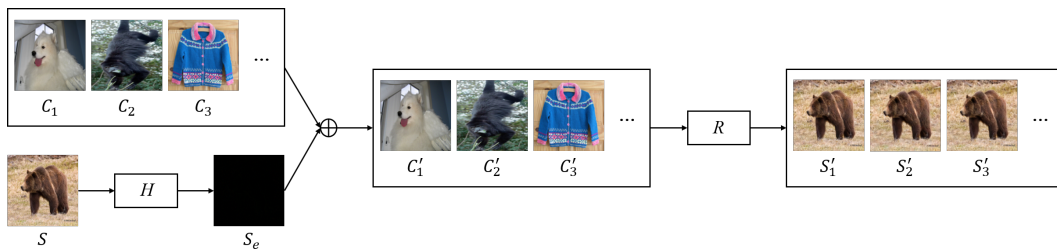

Figure 2: The proposed UDH meta-architecture: A secret image $S$ is fed to $H$ yielding $S_e$ which is added to a *random* cover image $C$ resulting in $C'$. Three example cover images are shown to demonstrate that $C$ can be any random natural image and has trivial influence on the revealed $S'$.

We propose a novel *(Universal) Deep Hiding* meta-architecture termed UDH as shown in Figure 2. Only the secret image is fed into $H$ and the encoded $S_e$ is added to a *random* cover image directly, *i.e.* $C' = C + S_e$. Note the similarity to adding a UAP to a random image in universal attacks [35, 49, 50, 6]. Different from the UAP to attack a target DNN, the universal $S_e$ is generated by co-training $H$ and $R$ to make it recoverable by $R$. The optimization goal is to minimize the loss defined as $\mathcal{L}(S, S_e, S') = ||S_e|| + \beta||S' - S||$, where $S_e = C' - C$ and following [2] we set $\beta$ to 0.75.

## 3.1 Basic setup and results

We co-train $H$ and $R$ on the ImageNet [13] training dataset with the ADAM optimizer [31]. The APD (average pixel discrepancy) performance evaluated on the ImageNet validation dataset is available in Table 1. The cover APD and secret APD are calculated as the $L_1$ norm of the difference between $C$ and $C'$ and that between $S$ and $S'$, respectively. Additionally, the results with Peak signal-to-noise ratio (PSNR), Structural Similarity (SSIM) and Perceptual Similarity (LPIPS) are reported. $H$ adopts a simplified U-Net from Cycle-GAN [56], and $R$ stacks several convolutional layers. The image resolution size is set to $128 \times 128$. Additional architecture details and results are provided in the supplementary. To compare with the existing DDH, we adopt a similar $H$ and $R$ and conduct the experiment under the same settings. Despite hiding images in a cover-agnostic manner, UDH achieves performance comparable to the existing DDH. Moreover, we empirically find that UDH leads to a more stable training (see the supplementary). Our result is comparable with the reported cover APD of 2.8/ 2.4 and secret APD of 3.6/ 3.4 in [2]/ [3]. We experimented with various architectures and found that the architecture choice for $H$ and $R$ has no significant influence on the performance. By design, UDH does not require a $P$ network, meanwhile for DDH, our exploration shows that adopting $P$ as in [2] does not provide superior performance and sometimes destabilizes training. The qualitative results of our UDH are shown in Figure 3, where identifying the difference between $C$ and $C'$ or that between $S$ and $S'$ is challenging. Note that their gap is amplified for better visualization.

Table 1: Performance comparison between UDH and DDH. The hiding and revealing performance are measured on the cover image $C$ and secret image $S$, respectively. For UDH $S$, we report two scenarios: one with $C'$ as the input of the $R$ network and the other with $S_e$ as its input. Higher is better for PSNR and SSIM, and lower is better for APD and LPIPS [54].

| Errors | APD↓ | PSNR ↑ | SSIM ↑ | LPIPS ↓ |
|---|---|---|---|---|
| UDH $C$ | 2.35 | 39.13 | 0.985 | 0.0001 |
| DDH $C$ | 2.68 | 35.87 | 0.977 | 0.0046 |
| UDH $S$ ($C'$) | 3.56 | 35.0 | 0.976 | 0.0136 |
| UDH $S$ ($S_e$) | 1.98 | 39.18 | 0.992 | 0.0022 |
| DDH $S$ | 3.50 | 34.72 | 0.981 | 0.0071 |

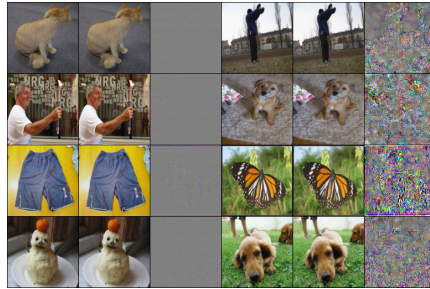

Figure 3: Qualitative results of UDH. The columns from left to right indicate $C$, $C'$, $S_e = C' - C$, $S$, $S'$, and $S' - S$ respectively.

**Remark on steganalysis.** We perform steganalysis on UDH. Resonating the findings for DDH in [2, 3], StegExpose [9], which detects LSB, is confirmed to fail for UDH while a DNN trained to detect secret information as a binary classifier can successfully detect the existence of hidden information. Prior works [2, 3] attribute this to the large hidden information capacity without providing further explanation. Our work provides intuitive explanation with visualization as well as understanding from the Fourier perspective.

# 4 Universal Deep Hiding analysis

**Where is the secret image encoded?** From $S$ to $S'$, the UDH pipeline performs two mappings, *i.e.* $H$ encodes $S$ to $S_e$ and $R$ decodes $S_e$ to $S'$. Since the APD between $S$ and $S'$ is very small, especially with $S_e$ as the input of $R$, the decoding can be seen as the inverse of the encoding. In the following, we analyse the encoding properties of UDH in the channel and spatial dimension.

We measure the channel-wise effect on $S_e$ and $S'$ by setting all values to zeros for a chosen channel in $S$ and $S_e$, respectively. The detailed results are shown in the supplementary. We observe that a change on any of the RGB channels in $S$ leads to similar APD values in all three channels in $S_e$, and the influence of $S_e$ on $S'$ mirrors the same behavior. The results indicate that the encoding mapping and decoding mapping are not channel-wise. With a similar procedure, we investigate the spatial dimension but set the pixel intensity of a single pixel to zero. Due to the local nature of the convolution operation, the influence is conjectured to be limited to only its surrounding pixels. We measure the APD with regard to the pixel distance from the point modified and report the results in the supplementary. We observe that for both encoding ($S$ on $S_e$) and decoding ($S_e$ on $S'$), the influence region is small. Our results align well with the findings in [2, 3], however, our more delicate analysis excludes the influence of $C$.

$S_e$ **visualization and Fourier analysis.** With the above analysis, it is clear that the secret image is encoded across all channels in channel dimension and locally in the spatial dimension; however, it is still not sufficient to understand the success of deep hiding. In Figure 4, we zoom into $S_e$ and visualize it together with its corresponding $S$. In the original image $S$, the pixel intensity values in the smooth region are the same or very similar, however, the corresponding values in $S_e$ are very different from its adjacent pixels, see zoomed patch 1 or patch 3. In particular, $S_e$ clearly shows a high-frequency (HF) property with repetitive patterns, different from natural images which mainly have low-frequency (LF) content. In the proposed UDH, the cover image $C$ can be perceived as a disturbance to $S_e$. It is intriguing that the decoding can work under such a large disturbance (note that the cover image is randomly chosen). The visualization results provide an intuitive explanation for its success. Since $R$ is implicitly trained to be only sensitive to HF content, adding a LF $C$ to $S_e$ barely corrupts the HF content of $S_e$, thus the disturbance of $C$ has limited influence. We further perform

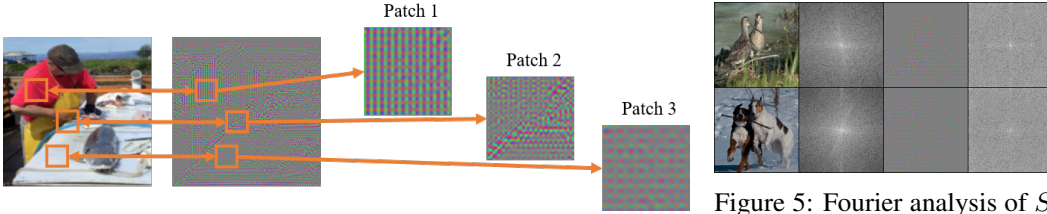

Figure 4: A sample secret image $S$ and its corresponding $S_e$. Three patches are zoomed for better visualization.

Figure 5: Fourier analysis of $S$ (left two columns) and $S_e$ (right two columns).

Fourier analysis of the natural images and $S_e$. The results are shown in Figure 5, which clearly shows that there is a clear frequency discrepancy between $C$ and $S_e$. We also conduct Fourier analysis for the result of hiding 3 secret images under the same cover (see Figure 10) and report the results in the supplementary. It shows that each $H$ network ends up using a different HF area in the Fourier space, which further suggests that frequency discrepancy is key for the success of deep steganography.

**Utilizing UDH to help visualize $S_e$ in DDH.** We have shown that $S_e$ in UDH mainly has HF content, which makes it robust to the disturbance of LF cover images. For the existing DDH, due to the cover dependence, we can not directly visualize $S_e$ or perform frequency analysis. However, we conjecture that $S$ is also encoded with a similar representation inside the $C'$ (not $S_e$ itself). The task to prove this conjecture is not trivial with only the existing DDH. Thus, we perform a cross-test for $H$ and $R$ from UDH and DDH. The output ($C'$) of $H$ of one meta-architecture is set as the input of $R$ of the other meta-architecture, and the results are shown in Figure 6. As expected, the revealed secret images $S'$ with $(H_u, R_u)$ and that of $(H_d, R_d)$ are similar. Note that the subscript "$d$" and "$u$" represent dependent and universal, respectively. Interestingly, at least for some images, the object shapes in $S'$ can still be clearly observed with the cross combination of $(H_d, R_u)$ or $(H_u, R_d)$. It shows that the secret image is also encoded with the same representation in $C'$ for DDH, otherwise it would be impossible for $(H_d, R_u)$ or $(H_u, R_d)$ to reveal any information about the secret image. Take $(H_d, R_u)$ for example, given that $R_u$ transforms HF content into LF content, $R_u$ would not be able to retrieve anything from $C'$ of $H_d$ if $H_d$ does not transform $S$ into HF content in $C'$ with similar representation of repetitive patterns.

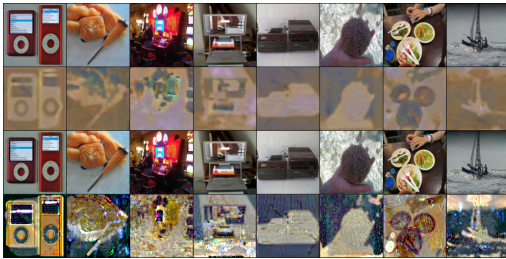

Figure 6: Cross-test with $H$ and $R$ from two different meta-architectures. The four rows from top to bottom indicate $S'$ with $(H_u, R_u)$, $(H_d, R_u)$, $(H_d, R_d)$ and $(H_u, R_d)$ respectively.

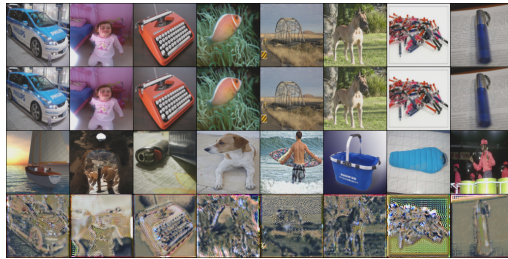

Figure 7: Analysis of the HF content in $C'$ for $R$ revealing the secret image. The four rows from top to bottom indicate $C'$, $C'$ with HF content filtered out, $S$ and revealed $S'$ with filtered $C'$.

To further verify that the DDH meta-architecture $R_d$ also transforms HF content in $C'$ to retrieve the secret image, we filter out the HF content in $C'$ for $(H_d, R_d)$ and the results are shown in Figure 7. It shows that filtering HF content in $C'$ leads to a total failure for the secret retrieval, confirming that indeed HF content in $C'$ is important for $R$ to reveal the secret image. We further experiment with retraining another $H_u$ to work in pair with a pretrained $R_d$ (fixed during the retraining). With no cover image imposed, the resulting secret APD is as small as 1.96,

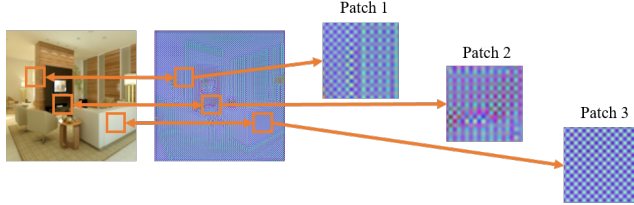

indicating that the new $H_u$ is equivalent to $H_d$ for pairing with the pretrained $R_d$. Since the new $S_e$ is independent of $C$, we visualize $H_u$ encoding in Figure 8. We observe a phenomenon similar to Figure 4, showing that DDH encodes the secret image into HF representation with repetitive patterns. Overall, our understanding of the success of deep steganography in UDH also helps explain how DDH works.

Figure 8: A secret image $S$ and its corresponding $S_e$ with zoomed patches for $H_u + R_d$ setup.

**Comparison of DDH and UDH.** For natural images, DDH and UDH achieve comparable performance as shown in Table 1. However, a difference between the frameworks arises when a pixel intensity change is applied.

DDH has the advantage that it can adapt the encoding of the secret image according to the cover image. For normal images, this property does not result in a significant performance difference. However, for a $C$ with a high amount of HF content, a performance difference between DDH and UDH can be observed due to the adaptive nature of the DDH framework. As

Table 2: Secret APD values when uniform random perturbations (magnitude varying from 10/255 to 50/255) are added to cover images.

| Arch | 10 | 20 | 30 | 40 | 50 |
|---|---|---|---|---|---|
| DDH | 3.3 | 3.7 | 4.3 | 5.0 | 5.9 |
| UDH | 10.6 | 21.5 | 33.0 | 43.8 | 52.3 |

Table 3: Secret APD values when different constant shifts (varying from 10/255 to 50/255) applied to container images.

| Arch | 10 | 20 | 30 | 40 | 50 |
|---|---|---|---|---|---|
| DDH | 7.8 | 13.7 | 21.0 | 27.0 | 32.4 |
| UDH | 3.5 | 3.5 | 3.5 | 3.5 | 3.5 |

shown in Table 2, with severe uniform random noise added to $C$, DDH is still able to recover the image with a low secret APD, while UDH fails in this context. The robustness of DDH to a noisy (HF) $C$ comes, however, at the cost of being sensitive to pixel intensity shift on the container image $C'$. The results in Table 3 show that with all pixel intensities of $C'$ shifted by a value of $50$, DDH can barely recover the secret image (APD: $32.4$), while the influence on UDH is not visible. This contrasting behavior can be attributed to the fact that the UDH framework by design trains $S_e$ to be robust to the disturbance of LF cover images, thus extra shift change, which is extremely LF, on $C'$ has limited influence. The robustness of UDH to pixel intensity shift on $C'$ makes it suitable for the application in LFM, see Sec. 5.3, because in general the light change is smooth. As an ablation study, we also report the results of (a) applying constant shift on $C$ or (b) applying uniform noise on $C'$ in the supplementary. (a) has negligible influence on DDH and UDH, while (b) leads to significant performance drop for both, but more for DDH.

## 5 Universal Deep Hiding applications

With the focus of hiding one full image, we apply UDH to steganography, watermarking, and light field messaging (LFM). Despite different goals, all of the three applications require the container image to look natural. Steganography has a focus of high hiding capacity, while watermarking and LFM prioritize robustness to distortions and light effects respectively. Steganagraphy also has the concern of evading steganalaysis, which is unlikely here due to large hiding capacity[3].

### 5.1 Universal deep steganography beyond hiding one image

**Flexible number of images for $S$ and $C$.** $S$ and $C$ are not required to have the same number of channels. We demonstrate the possibility of hiding $M$ secret images in $N$ cover images as well as hiding one or multiple color images in one gray image (Figure 9). Detailed results are shown in the supplementary. Without significant performance degradation, multiple $S$ can be hidden in one $C$,

and as expected, one $S$ can also be hidden in multiple $C$. The performance decreases when the task complexity increases, *i.e.* more $S$ and/or fewer $C$. Hiding $M$ images in $N$ cover images provides flexibility for practical hiding needs.

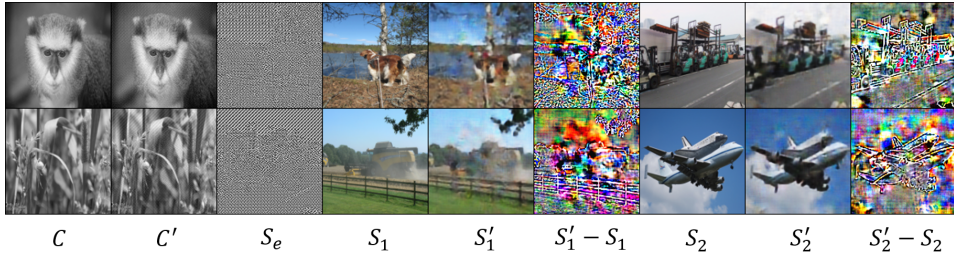

$$C \qquad C' \qquad S_e \qquad S_1 \qquad S_1' \qquad S_1'-S_1 \qquad S_2 \qquad S_2' \qquad S_2'-S_2$$

Figure 9: Hiding two color images in one gray image.

**Different recipients get different secret messages.** We experiment with multiple recipients receiving different $S$ images from the same $C'$. Similar to the proposed UDH in Figure 2, we train three pairs of $H$ and $R$ to encode and decode the corresponding secret images but hide the encoded secret content $S_{e1}$, $S_{e2}$, $S_{e3}$ in the same cover $C$, *i.e.* $C' = C + S_{e1} + S_{e2} + S_{e3}$. The overall procedure is demonstrated in Figure 10. More qualitative results are shown in the supplementary and

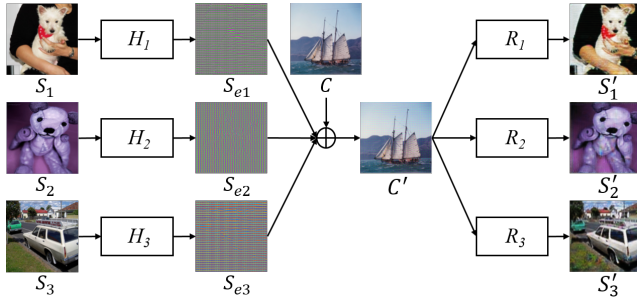

Figure 10: Pipeline for training multiple (3) pairs of $H$ and $R$ to hide 3 secret images under the same cover image.

we observe that the retrieving performance is reasonably good for all the three recipients ($R1$, $R2$, and $R3$) without revealing the wrong $S'$.

## 5.2 Universal deep watermarking

We apply the UDH to the task of watermarking. The primary advantage of watermarking with UDH is efficiency, *i.e.* requiring only one simple summation to watermark an image, which is especially meaningful in this era with vast amounts of images/videos. Watermarking with binary messages has been explored in HiDDeN [55], which can be seen as a special case of hiding images by treating barcodes as images. However, watermarking with images of a company logo, for instance, can be a more straightforward way to prove authorship.

Similar to [55], we analyze the robustness of UDH to various types of image distortions. Our method is by design robust to Crop and Cropout, however, we can only reveal the secret image hidden in the corresponding cropped area of the container image due to the spatially local property, see Sec. 4. To increase its robustness to dropout, Gaussian blurring, and JPEG compression, we train $H$ and $R$ on the relevant distortion and evaluate on the same type of distortion, and term them "specialized" model. Following [55], we also train a combined model that is robust to all of the above distortions.

**Watermarking by hiding images.** For all types of image distortions, we adopt the same parameter setting as in [55], except for JPEG compression [51] (see link[2] for more details). For making the model robust to various distortions, [55] adopts a single type of image distortion in the mini-batch for each iteration and swaps the type of adopted image distortion for a new iteration. In contrast, we divide the mini-batch equally into multiple groups, each group

Table 4: Secret APD performance with different image distortions. "Identity": training without distortions; "Specialized": training with a single corresponding distortion; "Combined": training with combined distortions.

| Model | Identity | Crop | Cropout | Dropout | Gaussian | JPEG |
|---|---|---|---|---|---|---|
| Identity | 3.5 | 5.5 | 6.0 | 42.5 | 53.2 | 57.0 |
| Specialized | 3.5 | - | - | 8.9 | 4.0 | 19.2 |
| Combined | 9.6 | 12.7 | 10.9 | 15.5 | 10.9 | 23.6 |

applying one type of image distortion. Empirically, we find that this simple change leads to faster

convergence and significantly improves the performance in our task. The results of evaluating model robustness are shown in Table 4. After training with combined image distortions, the model is found to be robust to all types of image distortions. The performance under JPEG compression is less favorable because JPEG mainly removes the HF information which is critical for the success of decoding the secret, see Sec. 4.

**Watermarking by hiding barcode.** A secret image has the content of $128 \times 128 \times 3$ bytes, while the binary information in [55] has 30 bits. The byte information can be seen as binary by transforming it into bit information through setting the pixel intensity lower than 128 as bit 0 and that higher than 128 as bit 1. With this transformation, the hiding capacity of UDH is still significantly higher than that in [55], *i.e.* $128 \times 128 \times 3$ bits vs. 30 bits. This significantly higher capacity comes from better utilization in the spatial dimension. To enable comparison with [55], we evaluate hiding pseudo-binary information, *i.e.* barcode, with the combined model trained for hiding an image. Note that retraining a specific model for hiding barcode might lead to higher performance. To demonstrate that our method is versatile, we intentionally avoid retraining. The pseudo-binary information is represented by dividing the secret image into $16 \times 16$ patches, each having the size of $8 \times 8 \times 3$. This pseudo-binary hiding is equivalent to hiding $16 \times 16$ bits information. As an ablation study, the performance of different patch size is

Table 5: Bits accuracy for the combined model under different distortions. Hiding more bits through decreasing patch size leads to lower retrieving accuracy.

| Patch Size | Total Bits | Identity | Dropout | Gaussian | JPEG |
|---|---|---|---|---|---|
| HiDDeN [55] | 30 | 100% | 93.0% | 96.0% | 63.0% |
| 2x2x3 | 4096 | 96.0% | 75.4% | 90.8% | 60.2% |
| 4x4x3 | 1024 | 99.9% | 92.7% | 99.5% | 73.4% |
| 8x8x3 | 256 | 100% | 99.6% | 100% | 91.5% |
| 16x16x3 | 64 | 100% | 100% | 100% | 99.4% |
| 32x32x3 | 16 | 100% | 100% | 100% | 100% |

also reported. Each patch has constant content of 0 or 255 to represent the bit value of 0 and 1 in the binary information, respectively. For the predicted output, we calculate the average value of each patch and classify the predicted bit output to 1 if the average value is higher than 128, otherwise 0. We observe that the bit accuracy decreases with smaller patch sizes, *i.e.* more hidden bits. The accuracy of our method in hiding 256 bits outperforms that of [55] in hiding 30 bits. For example, the accuracy of our approach under JPEG-50 is $91.5\%$ vs. their $63.0\%$. Qualitative results of the decoded barcode (or image) are shown in the supplementary. Due to large hiding capacity, empirically we find that some artifacts can be observed on the container image, which might be mitigated by retraining the model specifically for hiding barcodes or by adding adversarial learning as in [55].

## 5.3 Universal photographic steganography

Photographic steganography, also known as Light field messaging (LFM) [46], is the process of hiding and transmitting a secret message hidden in an image, displayed on a screen and captured with a camera. DNN based photographic steganography has been explored in [46]. The core difference between digital steganography and photographic steganography is that the latter one requires to transmit $C'$ from a display to a camera. This trans-

Table 6: Comparison of the generalization to unseen camera-display pairs. We compare the bit error rate (BER) of LFM [46] to the BER of the proposed UDH.

| Method | Setup A | Setup B | Avg. | LFM Avg [46]. |
|---|---|---|---|---|
| Frontal | 4.22% | 4.60% | 4.41% | 13.62% |
| 45° | 4.46% | 4.86% | 4.66% | 20.45% |

formation on $C'$ hinders the secret decoding with DDH [2]. To overcome this obstacle, [46] proposed to train a camera-display transfer function (CDTF) to cope with the distortion of the light field transfer. To train their CDTF function, they collected a dataset that contains more than 1 million images of 25 camera-display pairs, totaling 1.9TB. Given the size of their dataset, it is challenging to reproduce their results. Moreover, in their work, they show that the model performance decreases with a relatively large margin on an unseen camera-display pair. Given the aforementioned inherent robustness to $C'$ pixel intensity shift, UDH can work without the need of training a specific CDTF function. Following their procedure [46] applying homography to restore the image into a rectangular shape, we add a perspective transformation to the UDH training procedure to encourage invariance to such transformations. To not lose generality, the model is still trained to hide an image instead of a barcode [46]. We evaluate the trained model on commercial cameras (phones) and displays, and the performance is presented in Table 6. For the setup detail, refer to the supplementary. We observe that the average bit error rate (BER) is $4.41\%$, significantly lower than the average error of $13.62\%$ achieved by LFM [46]. For capturing the photo with an angle of $45°$, the performance

of [46] decreases by a large margin while our UDH is quite robust to such angle change. A concurrent work [43] based on DDH also solves this problem but involves various corruptions and a complex loss design. Note that our UDH training involves no additional corruptions except perspective transform and the loss is simply the same as defined as in Sec. 3. Moreover, our model is more versatile since it can also hide images, and the qualitative results are shown in Figure 11. Some artifacts can be observed on the decoded secret image, however, the performance is reasonable taking the task challenge into account. Our work is the first to achieve hiding an image for the task of LFM.

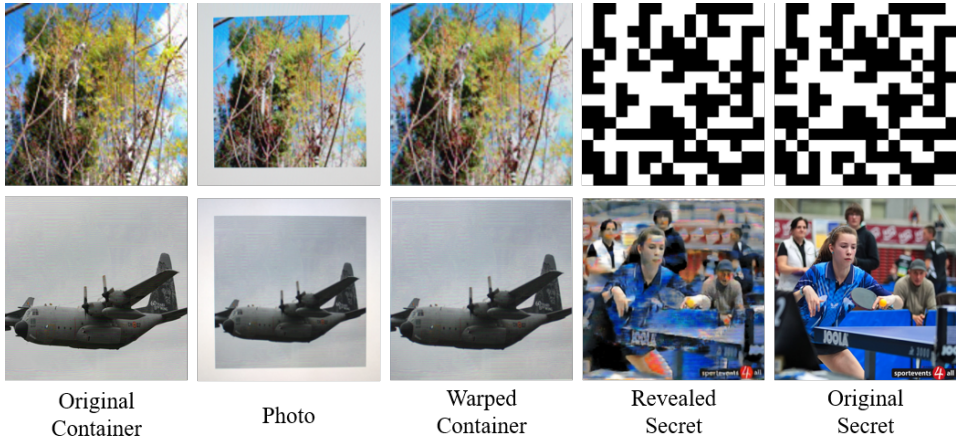

| Original Container | Photo | Warped Container | Revealed Secret | Original Secret |

Figure 11: Qualitative results of photographic steganography. The first row shows the example of hiding binary message, *i.e.* barcode, and the second row shows the possibility of hiding an image.

## 6   Conclusion

We proposed a novel deep hiding meta-architecture termed UDH, where $C$ behaves as disturbance and the encoding of $S$ is independent of $C$. Based on the proposed UDH, we analyzed where and how the $S$ is encoded, attributing the success of deep steganography to a frequency discrepancy between $S_e$ and $C$. Utilizing UDH also helps understand how DDH works. For deep steganography, beyond hiding one image in another, we demonstrated hiding $M$ images in $N$ images. We also showed that it is possible for different recipients to retrieve different secret images from the same $C'$. Exploiting the universal property of UDH, we applied it for efficient watermarking. In contrast to prior work only hiding binary information for watermarking, UDH can also hide images for watermarking. Applying UDH to LFM, UDH achieves state-of-the-art performance for hiding barcode. Moreover, with the LFM we successfully demonstrated transmitting an image with reasonable performance, opening the possibility of new applications for future work. Overall, our UDH is simple, effective yet versatile.

## 7   Broader impact

Information hiding is commonly used in an nefarious context, such as criminals secretly coordinating plans through messages hidden in images on public websites. However, we investigate the potential of deep hiding for beneficial applications. By comparing the existing DDH and the proposed UDH on various aspects, we provide an intuition behind the mechanisms of DNN-based deep hiding. With this understanding, we further push the simple use case of hiding one image in another to a more general case of hiding $M$ in $N$ images. Meanwhile, we demonstrate the possibility that different recipients can retrieve different secret images through the same container image, which can be used to provide different content to different users based on their practical needs. Intellectual property has become a major concern with the exponentially increasing number of images and videos. The proposed UDH constitutes a timely solution for addressing this issue with the concept of "universal watermarking". Finally, we show that UDH can be used for light field messaging. Different from prior works that only hide simple binary information, our work demonstrates the possibility of hiding a full image, which can greatly expand its use cases. For example, museums and exhibitions, can adopt light field messaging to provide a more informative and vivid experience for visitors.

## Acknowledgment

This work was supported by Deep Vision Farm (DVF).

## Footnotes

[2]Link: `https://github.com/ChaoningZhang/Pseudo-Differentiable-JPEG`

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
