[Supplementary Material]

# Supplementary for
# UDH: Universal Deep Hiding for Steganography, Watermarking, and Light Field Messaging

**Chaoning Zhang**[*]
KAIST
chaoningzhang1990@gmail.com

**Philipp Benz**[*]
KAIST
pbenz@kaist.ac.kr

**Adil Karjauv**[*]
KAIST
mikolez@gmail.com

**Geng Sun**
KAIST
tosungeng@gmail.com

**In So Kweon**
KAIST
iskweon77@kaist.ac.kr

This supplementary content is mainly organized in the order of being referenced in the main manuscript. To make it reader-friendly, for most of the sections here, we adopt the same titles as those in the main manuscript.

## A  Network architecture and training details

Table 1: $H$ architecture in DDH.

| Layer | input | channels | ks | stride |
|---|---|---|---|---|
| conv1 | $S$/$C$ | 6/64 | 4 | 2 |
| conv2 | conv1 | 64/128 | 4 | 2 |
| conv3 | conv2 | 128/256 | 4 | 2 |
| conv4 | conv3 | 256/512 | 4 | 2 |
| conv5 | conv4 | 512/512 | 4 | 2 |
| upconv5 | conv5 | 512/512 | 4 | 2 |
| upconv4 | conv4/upconv5 | 1024/256 | 4 | 2 |
| upconv3 | conv3/upconv4 | 512/128 | 4 | 2 |
| upconv2 | conv2/upconv3 | 256/64 | 4 | 2 |
| upconv1 | conv1/upconv2 | 128/3 | 4 | 2 |
| sigmoid | upconv1 | N/A | N/A | N/A |

Table 2: $H$ architecture in UDH.

| Layer | input | channels | ks | stride |
|---|---|---|---|---|
| conv1 | image $S$ | 3/64 | 4 | 2 |
| conv2 | conv1 | 64/128 | 4 | 2 |
| conv3 | conv2 | 128/256 | 4 | 2 |
| conv4 | conv3 | 256/512 | 4 | 2 |
| conv5 | conv4 | 512/512 | 4 | 2 |
| upconv5 | conv5 | 512/512 | 4 | 2 |
| upconv4 | conv4/upconv5 | 1024/256 | 4 | 2 |
| upconv3 | conv3/upconv4 | 512/128 | 4 | 2 |
| upconv2 | conv2/upconv3 | 256/64 | 4 | 2 |
| upconv1 | conv1/upconv2 | 128/3 | 4 | 2 |
| scale/255*tanh | upconv1 | N/A | N/A | N/A |

We adopt a simplified U-Net adopted in Cycle-GAN [6]. Specifically, we remove the two most inner convolutions and up-convolutions. The detailed $H$ architectures for the DDH and UDH are shown in Table 1 and Table 2, respectively, where the *conv* layer is followed by a BatchNorm layer and ReLU layer. In contrast to the final Sigmoid layer adopted in DDH, we adopt a Tanh layer multiplied by a scale factor which is set to $10/255$ by referencing the engineering choice in universal adversarial perturbations [3, 4, 1]. Note that different from [3, 4, 1], $S_e$ is minimized in the loss even with this constraint. With such a scale factor, some pixel intensities in $C'$ in UDH might still be out of the range $[0, 1]$. However, empirically we find that the

Table 3: $R$ architecture.

| Layer | input | channels | ks | stride |
|---|---|---|---|---|
| conv1 | image $C'$ | 3/64 | 3 | 1 |
| conv2 | conv1 | 64/128 | 3 | 1 |
| conv3 | conv2 | 128/256 | 3 | 1 |
| conv4 | conv3 | 256/128 | 3 | 1 |
| conv5 | conv4 | 128/64 | 3 | 1 |
| conv6 | conv5 | 64/3 | 3 | 1 |
| sigmoid | conv6 | N/A | N/A | N/A |

---

[*]Equal contribution

percentage of those pixels is very small, and limiting the range of $C'$ to $[0, 1]$ has an insignificant influence on the revealing performance. The architectures of the $R$ networks are shown in Table 3.

Different architectures, *i.e.* different depth for the $H$ network, with different image resolutions have been explored as well and the results are shown in Table 4.

Table 4: Performance comparison of different meta-architectures. We report the cover APD (cAPD) and the secret APD (sAPD), for which we report results with $C'$ (sAPD ($C'$)) and $S_e$ (sAPD ($S_e$)) as the input to $R$. "N/A" indicates that no secret image can be visually revealed resulting in a meaningless secret APD (higher than 60).

| meta-archs | cAPD | sAPD ($C'$) | sAPD ($S_e$) |
|---|---|---|---|
| full U-Net; image resolution of $256 \times 256$ | | | |
| DDH | 2.88 | 3.11 | N/A |
| UDH | 2.24 | 3.14 | 1.84 |
| simplified U-Net; image resolution of $256 \times 256$ | | | |
| DDH | 3.35 | 4.10 | N/A |
| UDH | 2.33 | 3.65 | 2.19 |
| simplified U-Net; image resolution of $128 \times 128$ | | | |
| DDH (with $P$) | 6.42 | 5.26 | N/A |
| DDH | 2.68 | 3.50 | N/A |
| Universal | 2.35 | 3.56 | 1.98 |

**Training curve.** We note that the proposed UDH achieves comparable performance than the existing DDH meta-architectures after training for 60 epochs with the initial learning rate of 0.001 (decay by a factor of 10 at epoch 30). The training curve is shown in Figure 1. We observe that the training curve of our UDH is much more smooth, which might be attributed to the point that $H$ encodes $S$ independent of $C$. The DDH encoding process is dependent on $C$, thus it might be over-fitting for some certain $C$ and makes it relatively less generalizable. The smooth training of our UDH might also provide another reason why without exploiting the knowledge of $C$, our UDH achieves comparable performance with the existing DDH.

Figure 1: Training curves for UDH and DDH.

# B    Universal deep hiding analysis

## B.1    Where is the secret image encoded?

The results of investigating where the secret image is encoded for channel and spatial dimension are shown in Table 5 and Figure 2, respectively.

Table 5: Influence of $S$ on $S_e$ (left) and that of $S_e$ on $S'$ (right) by setting one channel in $S$ and $S_e$, respectively, to zero values.

| | R ($S$) | G ($S$) | B ($S$) |
|---|---|---|---|
| R ($S_e$) | 2.27 | 2.55 | 2.82 |
| G ($S_e$) | 2.62 | 3.84 | 1.78 |
| B ($S_e$) | 2.62 | 3.04 | 2.15 |

| | R ($S_e$) | G ($S_e$) | B ($S_e$) |
|---|---|---|---|
| R ($S'$) | 16.78 | 41.01 | 17.30 |
| G ($S'$) | 23.19 | 35.56 | 10.18 |
| B ($S'$) | 34.39 | 25.39 | 13.79 |

Figure 2: Influence of setting one pixel intensity value to zero. Influence of $S$ on $S_e$ (left), $S_e$ on $S'$ (middle), $S$ on $S'$ (right). The influence is measured with the APD with regard to the distance to the modified pixel.

**Is every channel equally important?** we have demonstrated that the secret image is encoded across all channels by showing that change on a single channel in $S$ (or $S_e$) has an almost equivalent influence on the three channels in $S_e$ (or $S'$). However, it remains yet to know whether every channel in $S_e$ is equally important for revealing $S'$. To this end, we explore the robustness of $S_e$ to channel dropout, *i.e.* whether the secret image can be revealed with only partial channels of $S_e$ and the results are shown in Figure 3. We observe that the revealed $S'$ are well recognizable with any two channels (see the bottom three rows of Figure 3). With only one channel, the secret image can still be revealed to some extent (see the second to the fourth row of Figure 3). Not every channel is equally important. Specifically, we find that channel $G$ carries the most information and the $B$ channel carries the least information (compare the second to fourth rows of Figure 3). The algorithm automatically chooses to focus more on the $G$ channel without any prior constraint in the training. With only one channel $G$, the revealed $S'$ looks visually similar to that with all channels, but with a significant color change. Since one channel can carry much of the information of $S$, it is not surprising that hiding one or multiple color image(s) in a gray image is possible.

Figure 3: Robustness of $S_e$ to channel dropout. The first row indicates $S'$ with all three channels (r,g,b)=$(1, 1, 1)$. The following three rows indicate using only one channel of $S_e$, with (r,g,b)=$(1, 0, 0)$, $(0, 1, 0)$ and $(0, 0, 1)$, respectively. The last three rows indicate using only two channels of $S_e$, with (r,g,b)=$(0, 1, 1)$, $(1, 0, 1)$ and $(1, 1, 0)$, respectively. "1" indicates the corresponding channel of $S_e$ remains unchanged, "0" indicates the corresponding channel of $S_e$ is set to zero.

## B.2 Understanding the existing DDH meta-architecture

In the main manuscript, we performed a cross-test with $H$ and $R$ from two different meta-architectures, *i.e.* DDH and UDH but without retraining. Here, we fix the weight of $R$ but retrain a new $H$ from a different meta-architecture to work in pair with the pre-trained $R$. The results are shown in Table 6. We observe that a new $H_d$ can be easily trained to work with a pre-trained $R_u$. However, it is much more challenging to train a new $H_u$ to work with a pre-trained $R_d$ when the cover image $C$ is present. This significant performance drop is mainly caused by the significant disturbance of $C$. As an ablation study, we repeat the same procedure but exclude $C$. The secret images can be revealed almost perfectly with secret APD as low as 1.96.

Table 6: Cross-test results by training a new a new $H$ from a different meta-architecture to work in pair with a pretrained $R$. The subscripts $d$ and $u$ indicate DDH and UDH, respectively.

| Architecture | cAPD | sAPD |
|---|---|---|
| $H_d + R_u$ | 3.79 | 3.82 |
| $H_u + R_d$ | 3.66 | 18.28 |
| $H_u + R_d$ (w/o cover) | 0.48 | 1.96 |

Table 7: Secret APD values when different constant shifts (varying from 10/255 to 50/255) applied to cover images.

| Arch | 10 | 20 | 30 | 40 | 50 |
|---|---|---|---|---|---|
| DDH | 3.6 | 3.7 | 3.8 | 4.0 | 4.1 |
| UDH | 3.5 | 3.6 | 3.6 | 3.6 | 3.7 |

Table 8: Secret APD values when uniform random perturbations (magnitude varying from 10/255 to 50/255) are added to container images.

| Arch | 10 | 20 | 30 | 40 | 50 |
|---|---|---|---|---|---|
| DDH | 30.1 | 54.7 | 71.9 | 86.1 | 96.0 |
| UDH | 10.9 | 21.9 | 33.0 | 43.7 | 51.7 |

## B.3 Comparison of DDH and UDH

Applying constant shift to the cover images $C$ (Table 7) has limited influence on DDH and UDH. Adding random noise to container image $C'$ (Table 8) degrades performance on DDH and UDH, but more on DDH. Together with Tables 3 from the main manuscript, we conclude that UDH is more robust to corruptions on $C'$.

# C Universal deep hiding applications

## C.1 Beyond hiding one image in one image

**Hiding $M$ images in $N$ images.** Quantitative results are shown in Table 9. The results of hiding images with different resolutions and channels are shown in Table 10. The qualitative results for hiding 6 color images in 3 color images are shown in Figure 4. The results show that $C'_i$ looks individually similar to $C_i$, and $S'_i$ is also indistinguishable from $S_i$, which indicates the success of hiding multiple (6) in multiple (3) images. Overall, in the above experiments, the APD for both cover and secret images is not significant and a human observer can not observe obvious visual differences.

Table 9: APD for hiding M secret images in N cover images.

| index | N (cover) | M (secret) | cover APD | secret APD |
|---|---|---|---|---|
| 1 | 3 | 1 | 1.83 | 3.05 |
| 2 | 1 | 3 | 3.42 | 6.74 |
| 3 | 1 | 4 | 3.83 | 8.63 |
| 4 | 3 | 3 | 3.38 | 5.94 |
| 5 | 3 | 6 | 3.82 | 8.41 |
| 6 | 3 | 12 | 4.51 | 13.79 |

**Different recipients get different secret messages.** The pipeline of training multiple pairs of $H_i$ and $R_i$ to hide multiple secret images in the same cover is shown in the main manuscript. $H$ and $R$ work in pairs and each $R$ can only reveal the secret message encoded by its corresponding $H$. Note that each pair of $H$ and $R$ works independently, *i.e.* during the inference stage excluding other pairs of $H$ and $R$ will not influence the working mechanism of the left pair. Intuitively, for those

Table 10: APD for hiding images in images with different resolutions and channels. For each entity, the first value indicates the resolution and the second one indicates the channel. For the channel, 1 indicates a gray image, 3 indicates a single color image, 6 indicates 2 color images.

| index | cover | secret | cover APD | secret APD |
|-------|-------|--------|-----------|------------|
| 7 | 64/ 3 | 128/ 3 | 2.93 | 5.94 |
| 8 | 32/ 3 | 128/ 3 | 3.58 | 9.56 |
| 9 | 128/ 1 | 128/ 3 | 4.67 | 7.73 |
| 10 | 128/ 1 | 128/ 6 | 5.76 | 10.63 |

Figure 4: Results for hiding 6 secret images in 3 cover images. The first 6 rows indicate the three cover images and their corresponding container images. The following 12 rows indicate the 6 secret images and corresponding revealed images.

pairs of $H$ and $R$ to work independently with little influence on each other, each $S_e$ needs to be encoded differently. To analyze this, we feed the same image into the three $H$ networks to exclude the influence of the images. Example cover ($C$), secret images ($S_1, S_2, S_3$), containers ($C'$), and revealed secrets ($S_1', S_2', S_3'$) are shown in Figure 5. The results for the visualization and Fourier analysis of $S_{e1}, S_{e2}, S_{e3}$ are shown in Figure 6. First, it can be observed that $S_e$ generated by the same $H$ have very similar frequency patterns. However, (for the same image) $S_e$ generated by different $H$ have different patterns. Especially, the frequency pattern for $S_3$ is very different from those of $S_1$ and $S_2$. The frequency discrepancy between $S_1$ and $S_2$ is less significant but still observable. The success of this pipeline is that each $R$ has been trained to be sensitive to $S_e$ of only a certain frequency type and treats the cover image $C$ as well as $S_e$ of other frequency types as noise. Carefully comparing the visualization results of different $S_e$, we note that $S_{e1}$ and $S_{e2}$ mainly have patterns repeated in vertical and horizontal directions respectively, while $S_{e3}$ has clear repetitive patterns in both horizontal and vertical directions.

| $C$ | $C'$ | $S_e$ | $S_{e1}$ | $S_{e2}$ | $S_{e3}$ | $S_1$ | $S_2$ | $S_3$ | $S_1'$ | $S_2'$ | $S_3'$ |

Figure 5: Hiding multiple secret images under one cover and different recipients, *i.e.* $R$ networks, reveal different secret images without influencing each other.

Figure 6: Visualization and Fourier analysis of $S_{e1}, S_{e2}, S_{e3}$. We intentionally choose the same secret images for them to exclude the influence of the image but analyze the difference of $H_1$, $H_2$ and $H_3$. The top 4 rows indicate the secret images, $S_{e1}, S_{e2}, S_{e3}$, respectively. The bottom 4 rows indicate their corresponding Fourier analysis.

One application of this technique is to fool a third party that tries to steal $R$ for revealing the secret message. $R$ is needed for revealing the secret message, however, a wrong $R$ can be intentionally

leaked to a third party to let it retrieve the wrong message. This result suggests that aiming to steal $R$ can be dangerous for revealing the intentionally misleading message.

## C.2 Universal deep watermarking

Figure 7: Secret revealing performance without special training. The first row indicates original secret images, the following rows indicate revealed secret images with different distortions in the order: crop, cropout, dropout, Gaussian blurring, and JPEG compression.

Figure 8: Secret revealing performance when UDH is trained with (and only with) relevant image distortion. The first row indicates original secret images, the following rows indicate revealed secret images with different distortions in the order: dropout, Gaussian blurring, and JPEG compression.

**Watermarking under specialized distortion.** The results under various types of distortion for a normally trained $H$ and $R$ without adopting the corresponding distortion in the training are shown in Figure 7. We observe that our approach is robust to crop and cropout but not robust to dropout, Gaussian blurring, and JPEG compression. To verify the effectiveness of adopting the corresponding distortion in the training, we visualize the respective results in Figure 8. We note that the secret image can be revealed almost perfectly if the model is specifically trained only for that distortion.

**Watermarking with combined robustness.** In the main manuscript, we have discussed that hiding binary information can be seen as a special case of our exploration. The comparison with [5] shows that our approach can hide more information while achieving higher revealing accuracy. The qualitative results are shown in Figure 9. We further visualize the decoded *barcode* and *images* under different distortions in Figure 10 and Figure 11 respectively.

Figure 9: Results of hiding 256 pseudo-bits (each bit being represented as an $8 \times 8 \times 3$ block) in images. The top three rows are cover, container images, and their gap respectively; the bottom three rows are secret bits, revealed secret bits and their gap, respectively.

## C.3 Universal photographic steganography

**Setups for the Photographic Steganography results.** In the main manuscript, we considered two setups for Photographic Steganography. Setup A is for a display-DSLR pair, while setup B is for a display-cellphone pair. We tried with numerous images with different display-camera (display-cellphone) pairs. We observe that the choice of display or camera/cellphone has trivial influence on the revealing performance. It is somewhat expected since our approach is not trained on any hardware setup as in [2], thus avoiding the issue of over-fitting to any specific hardware setup. In our setup, for consistency, the image resolution is set to $128 \times 128$ with the number of the hidden bit set to 256. It is equivalent to [2] which hides 1024 bits in an image of resolution $256 \times 256$. Additional qualitative results for photographic steganography are shown in Figure 12.

Figure 10: Qualitative results of revealing 256 pseudo-bits (each bit being represented as an $8 \times 8 \times 3$ block) under different distortions for a combined model.

| Original | Identity | Dropout | Gaussian | JPEG |

Figure 11: Qualitative results of revealing images under different distortions for a combined model. For the last row (NeurIPS logo), the 512x512 image resolution is used for better visualization.

| Original Container | Photo | Warped Container | Revealed Secret | Original Secret |

Figure 12: Additional photographic steganography results. The first three rows correspond to hiding a binary message (barcode), while the last three rows demonstrate the image hiding.