[Reviews · NeurIPS 2020]

Review 1

Summary and Contributions: This paper presents a method for “universal” image steganography, where the secret is encoded without any knowledge of the cover image. Many different experiments are performed to demonstrate the relative performance of this to “data-dependent” steganography, the effects of changing bits in the message, the spectral properties of the hidden message residual, and comparisons to other deep learning methods in the context of watermarking and photographic steganography.

Strengths: This paper contains a very large number of different experiments and analyses, with lots of results on both digital and photographic transmission. In the context of this framework, there is only a minor loss in performance when going from the data-dependent DDH to the universal UDH, and UDH provides a non-negligible increase in training stability and allows for more efficient embedding of the same message into many cover images. The paper draws a nice analogy between UDH and work on universal adversarial perturbations. The experiments demonstrate that the proposed UDH method outperforms HiDDeN for robust watermarking while hiding many more bits. The ability to encode multiple messages to multiple recipients in one C’ is interesting and is a natural extension of this framework; it’s hard to imagine this working as well with cover-dependent hiding. The analysis in the supplement along with Figure 12 is particularly intriguing, showing how each H network ends up using a different high frequency subset of Fourier space. The recovered secret image shown in Figure 18 of the supplement is impressive for photographic steganography. I would be interested to hear more details about the “perspective transform” added to the training procedure here and whether any other distortions were applied during training, and also how the DDH version compares in the setting of Table 7.

Weaknesses: It seems like the DDH framework does not use a residual network, which could have a big impact on performance. Directly outputting an image versus outputting a residual to be added to the input each lead to drastically different behavior (for example, in the case of image denoising, switching to a residual network immediately provides a large performance boost). *If* the DDH network is not outputting a residual and is instead directly outputting C’, I am suspicious of the comparison being done here. One example of this affecting the paper’s results is Table 6: if the DDH network output a residual, it would be just as robust to constant shifts as the UDH network. Additionally, it would be easier to do the same type of Fourier analysis as in Figure 6 if the DDH network output a residual: you could simply produce a bunch of cover-dependent residuals for the same secret and look at their aggregated Fourier domain statistics. I am unsure why the distortions in section 5 are not directly combined during training. It sounds like each subgroup of the minibatch only has one distortion applied, as opposed to StegaStamp [23], where the corruptions are randomly combined for every training example, or LFM [25], where the corruption of the entire pipeline is simulated by a network. It seems like this might confer increased robustness rather than only ever using one corruption at once during training. (I am unclear if this is what L125-136 in the supplement is talking about or not?) Comparing to LFM in Table 7 with a different setup seems a bit unfair, I’m not sure if this would be fair without being able to capture and decoded LFM images in the exact same setup. (I realize this is not easy to do, and the numbers for this method still are impressive in absolute without comparing them to LFM.)

Correctness: As far as I can tell, everything is correct.

Clarity: Each individual part is well written. However, there are so many different sections and experiments that it might be beneficial to have a brief enumeration at the end of the introduction listing them. The supplement could certainly benefit from a table of contents since it also contains a lot of additional experiments.

Relation to Prior Work: Related work on deep steganography is covered well and as far as I know this is the first work to introduce "universal" deep steganography with an encoding that does not depend on the cover image.

Reproducibility: Yes

Additional Feedback: It would be easier to interpret the image results if PSNR was used instead of APD in all tables. (That may be a matter of personal preference though.) ===================== Post-rebuttal feedback: Given the additional information provided in the rebuttal, it seems like the benefit is not only coming from the residual network structure. In light of this, I have increased my score.


Review 2

Summary and Contributions: In this paper, the authors propose a novel universal deep hiding secret images in cover images. In this framework, the secret image is transformed first and then added to the cover image. The authors also analyze their proposed method and find that the frequency discrepancy contributes to the success of the proposed method. The proposed method can achieve state-of-the-art performance and can perform hiding multiple images in multiple images.

Strengths: - A deep learning based universal steganography method that is not dependent on the cover data. Extensive experiments show the proposed method has stable performance in hiding meta information such as an image or binary data such as watermarking. - The proposed method fascinates the analysis of the encoded content. The visualization of the frequency domain of the encoded secret image proves the frequency discrepancy. The analysis contributes to the understanding of deep steganography and further improve the results. - The demonstration of photographic steganography seems promising. - The method they proposed outperforms state-of-the-art methods. - The authors conducted enough analysis of encoded images via visualization and frequency analysis that gives good insights on how the networks hide information into images. - The authors showed the robustness to modification and steganalysis of their method, which is important in a real-world digital watermarking scenario.

Weaknesses: - In section 5, using JPEG compression without backpropagation seems questionable. As in this way, the encoder will not adjust the encoding methods to protect the information from the compression. In section 4.2, high-frequency information is significant for recovering the secret data. However, JPEG is known to clip the high-frequency information. If the encoder is not adjusted, the conclusion is kind of contradictory. - In section 4.1, the analysis of spatial and channel dimension is not convincing enough. Is there the possibility it depends more on the deep architecture or the training methods.

Correctness: In general, Yes. It is claimed that the model needs relatively small data than other methods, but there are no explicit support materials for this argument.

Clarity: The paper is well written in general. One question: It seems the input size of images is fixed (not sure because of the lack of enough description on networks.). It will lead to limited applications on real-world scenarios.

Relation to Prior Work: Yes,

Reproducibility: Yes

Additional Feedback:


Review 3

Summary and Contributions: ================= Post rebuttal ======================== I have read the reviews of the fellow reviewers and the response of the authors. The major concern as per R4 against this paper is the use of the term deep steganography when the proposed method is not robust to steganalysis. The authors have clarified the use of this term, and keeping in mind the original goals of this field of study, have proposed to change the term to deep hiding. Besides this confusion, I believe the work is still worthy of being presented at NeurIPS. My reasons are as follows - (a) the authors have well explained the motivation of the work, which is to move beyond data dependent image hiding and explore universal hiding schemes. (b) They have proposed a method that is able to do, thus opening up the area to new possibilities. Through a series of experiments (quantitative and qualitative) the authors have compared DDH and UDH on various aspects and provided intuition behind how deep nets attempt a solution to the problem of merging and separating images C and S. (c) They have discussed the limits of this approach when it comes to steganalysis i.e. where it works and where it fails. Thus, while they offer to the community a wholesome understanding and analysis of the solution(s) till date, they also present what open challenges still exist. I would retain my score of accepting this paper. =================================================== This paper proposes a deep neural model for universal data hiding i.e. hiding of a secret image (S) in any given cover image (C), by encoding it as S_e = H(C), and decoding it as S' = R(C+S_e), such that C+S_e is perceptually the same as C. This contrasts with the existing DDH (dependent data hiding) where the encoded image is a function of both S and C. Their work draws motivation from the successful discovery of universal adversarial perturbations (attack perturbations that can cause a large fraction of input images to be misclassified without being perceptible). The authors are able to demonstrate that UDH is indeed possible and that it works at par with DDH. This further simplifies the analysis of the encoder as a function of S alone, allowing it to be interpreted via a qualitative comparison of S and S_e for similarities and differences.

Strengths: 1) The paper is well-written and the related work seems thorough. 2) The experiments are multi-faceted and insightful. Particularly, the three: a) Frequency analysis of natural images versus encoded images (qualitatively, fig. 6) b) Corruption of the encoding and decoding by adding HF content to C (using of synthetic versus noise images as C) c) Cross encoding and decoding (using the encoder of UDH with decoder of DDH, and vice-versa).

Weaknesses: 1) Presentation is cluttered, with a lot of important results put in the appendix. For eg. ''b) Corruption of the encoding and decoding by adding HF content to C (using of synthetic versus noise images as C)''

Correctness: Yes, they seem to be.

Clarity: Yes.

Relation to Prior Work: Yes.

Reproducibility: Yes

Additional Feedback: 1) Since the Analysis of UDS is core to the work, perhaps Sec.4. could be pushed ahead of discussion on variations of the vanilla setup. 2) The choice of perturbations in Table 5 and Table 6 seem a bit arbitrary. What would happen to the results in Table 5 and Table 6 if the perturbations are reversed, i.e. a constant shift is applied to the cover mages and uniform random perturbations to the container images. More particularly,to support their hypothesis, would it be possible to see how S_e changes with constant shift added to C in DDH model? Let's say the changed encoding is S'_e. We should then be able to see that (C+S_e) + constant_shift != (C+constant_shift) + S'_e. Any comments? 3) The images could be made bigger in the interest of readability.


Review 4

Summary and Contributions: The authors propose a novel universal deep hiding (UDH) meta-architecture to disentangle the cover image and secret image. They present efficient watermarking and their method outperforms HiDDeN.

Strengths: They construct rich experiments and experimental performance is good. The paper analyzes the hiding data by location and frequency. Frequency analysis is a contribution to the understanding of DNNs for hiding data.

Weaknesses: The reviewer believes that the paper is seriously flawed and should not be published in NeurIPS. 1. The incorrect understandings of steganography and watermarking lead to wrong problem definition and the misleading experiments and conclusions. The authors claim that they sacrifice secrecy for large hiding capacity. Actually, the challenge of steganography is not hiding data but deceiving steganalysis. Hiding large data is far simpler than deceiving the steganalysis. Steganography fails when it does not consider fighting against steganalysis. The method for hiding and recovering data is not called steganography but is in other fields. Therefore, the core idea of the paper is incorrect. The reviewer recomend the authors to research more on steganography [1-7]. 2. The proposed UDP is not described clearly. The authors only provide Line 97-109 to describe their method, which is not enough for the reviewer to understand the proposed UDH. 3. Lack of novelty. The proposed methodology is similar to [8]. Sect.3.2 has been explored by [9]. It seems that the performance of [9] outperforms this paper. [1-7] have researched on steganography deeply. This paper analyzes the simple frequency phenomenon instead of deeper reasons. [1] V. Holub, J. Fridrich, and T. Denemark, “Universal distortion function for steganography in an arbitrary domain,” EURASIP Journal on Information Security, vol. 2014, no. 1, pp. 1–13, 2014. [2] B. Li, M. Wang, J. Huang, and X. Li, “A new cost function for spatial image steganography,” in Proc. IEEE 2014 International Conference on Image Processing, (ICIP’2014), 2014, pp. 4206–4210. [3] V. Sedighi, R. Cogranne, and J. Fridrich, “Content-adaptive steganography by minimizing statistical detectability,” IEEE Transactions on Information Forensics and Security, vol. 11, no. 2, pp. 221–234, 2016. [4] J. Fridrich and T. Filler, “Practical methods for minimizing embedding impact in steganography,” in Proc. SPIE, Electronic Imaging, Security, Steganography, and Watermarking of Multimedia Contents IX, vol. 6505, 2007, pp. 650 502–1–650 502–15. [5] T. Denemark and J. Fridrich, “Improving steganographic security by synchronizing the selection channel,” in Proc. 3rd ACM Information Hiding and Multimedia Security Workshop (IH&MMSec’ 2015), 2015, pp. 5–14. [6] B. Li, M. Wang, X. Li, S. Tan, and J. Huang, “A strategy of clustering modification directions in spatial image steganography,” IEEE Transactions on Information Forensics and Security, vol. 10, no. 9, pp. 1905– 1917, 2015. [7] W. Tang, B. Li, W. Luo, and J. Huang, “Clustering steganographic modification directions for color components,” IEEE Signal Processing Letters, vol. 23, no. 2, pp. 197–201, 2016. [8] Shumeet Baluja. Hiding images in plain sight: Deep steganography. [9] Shumeet Baluja. Hiding images within images.

Correctness: The reviewer believes that the claims of the authors have serious flaws. Details can be found in ''Weakness''.

Clarity: The paper is not well written. Because there are many claims are not convincing. The proposed method is not described clearly and the motivation for the experiments is confusing.

Relation to Prior Work: The reviewer believes that the authors did not research clearly on steganography. There are plenty of awesome papers for explaining how steganography and watermarking work.

Reproducibility: Yes

Additional Feedback: Make a major revision and consider other core ideas of the paper. The method for analyzing is interesting.

[Author Response · NeurIPS 2020]

We thank all reviewers for their insightful comments. We will improve readability and structure as suggested by R1 and R3.

**[R1]**: **Residual DDH (R-DDH).** As suggested, R-DHH is explored and indeed outperforms DDH but only by a small margin (R-DHH 2.36/3.46 vs. DHH 2.68/3.50 for cAPD/sAPD). The corresponding results in Table 6 for R-DDH are 7.2/12.8/19.5/24.2/29.1 for a shift of 10/20/30/40/50. At first sight, the UDH robustness to constant-shift corruptions might be attributed to its "residual" property. We found that the shortcut helps R-DDH behave like UDH in the early stage of training, however, with both $C$ and $S$ as the input of network $H$, $H$ eventually learns to adapt to encode $S$ dependent on $C$. To further prove this, we add the residual $S_e$ in the R-DDH to a random cover, no $S$ can be revealed. Such encoding dependence is due to the $C$ shortcut in R-DHH facilitates training instead of behaving like an independent "cover noise" as in UDH, highlighting the importance of universal property of UDH.

**Corruption for watermarking.** In Sec. 5, we compare with HiDDeN[29] which randomly adds one corruption type per mini-batch and evaluates all corruptions separately. Thus, we mimic HiDDeN but adopted our dividing strategy instead of their swap strategy, resulting in a significant performance boost as shown in Table 3. Note [23,25] are for the task of LFM not watermarking.

**Concerns regarding LFM task.** Different from StegaStamp[23] applying various corruptions and a delicate loss design, we only used perspective warp[1] in training with a simple loss described in lines 94-95. Despite the simplicity, UDH achieves impressive performance as pointed out by R1. We confirmed that DDH fails in the setting of Table7, which is consistent with the Baluja result in Figure 2 of LFM[25]. Training a CDTF network as in [25] requires a 1.9TB display-camera pair dataset and finally leads to over-fitting, resulting in a performance drop for unseen display-camera pairs. Excluding the need for such collected display-camera data, our approach is not over-fitting to any certain type of hardware devices, thus the comparison in Table7 is fair to a large extent. Additional new comparison with [23] shows a mean accuracy for our UDH and [23] is $98.8\%$ & $97.6\%$, respectively for hiding 100 bits in $256\times256$ under our hardware setups. During evaluation, [23] is not flexible with changing #bits, while ours is flexible and also versatile for hiding image.

**[R2]**: **JPEG compression.** In essence, the JPEG compressed container image is simulated by only adding a "noise" to the original container image. Since this "noise" is $JPEG(C') - C'$, and has the "JPEG compression" pattern, the pipeline ($H$ and $R$) would be naturally adapted to be robust to it even though it is only added as "noise" without back-propagation. Table3 supports the effectiveness of training with our JPEG with an APD of 23.6 vs. 60.1 (Mask[29]) and 58.5 (Drop[29]). Indeed, JPEG suppresses HF content and leads to a performance drop (9.6 vs. 23.6 see Table3) even with such adaptation in training. The Fourier analysis on the right reveals a distinctive, different pattern for training without JPEG augmentation (top) from that with it (bottom). **Architecture & training method influence.** We apply U-Net and ResNet of 3 sizes for $H$ and $R$, experiment with SGD, ADAM, and decreasing and cyclic learning rate. All variants show a consistency with the results in Sec. 4.1. **Data efficiency.** We clarify that the data efficiency claim is made for the task of LFM (see line310-314). Compared to [25], UDH is data-efficient in the sense that arduous collection of a separate (1.9TB) dataset is not necessary, since UDH can achieve its robustness by training on ImageNet. **Image size.** For both DDH and UDH, the image size during evaluation is flexible. We confirmed the results in Table 1 are equivalent for image size ranging from 64, 128, 256 to 512.

**[R3]**: **Choice of perturbations in Table 5 & 6.** Applying constant shift to the cover images $C$ (Table Left) has limited influence on DDH and UDH. Adding random noise to container image $C'$ (Table Right) degrades performance on DDH and UDH but more on DDH. Together With Table 5 & 6, we conclude:

| Arch | 10 | 30 | 50 | | Arch | 10 | 30 | 50 |
|------|-----|-----|-----|---|------|------|------|------|
| DDH | 3.6 | 3.8 | 4.1 | | DDH | 30.1 | 71.9 | 96.0 |
| UDH | 3.5 | 3.6 | 3.7 | | UDH | 10.9 | 33.0 | 51.7 |

UDH is more robust to corruptions on $C'$, while DDH is more robust to HF corruptions on $C$. As suggested, for DDH we visualize the difference between $S_e$ and encoding $S_e'$ with a constant shift on $C$ (See image below). $S_e$ and $S_e'$ are different, naturally leading to $(C + S_e) + shift \neq (C + shift) + S_e'$ explaining why DDH is not robust to constant shift on container image $C'$.

**[R4]**: **Traditional Steg vs. Deep Steg.** Steganalysis is indeed a major concern for Steg (steganography), but hiding a large data capacity is also non-trivial. According to Baluja[1], traditional Steg methods, e.g. HUGO, have a small hiding capacity of <0.5 bpp, while their DDH hides a full image (24bpp). Different from traditional Steg targeting accurate bit information, deep steg[1,24,26] hides an image (technically byte information) and loosely recovers it with the goal of less distortion on $C$. Our work attempts understanding the success of *Deep Steg* hiding an image. We appreciate R4's suggested 7 papers related to understanding *traditional Steg* and will cite them to clearly differentiate our work.

$C$      $S$      $S_{e\ (shift=0)}$   $S'_{e\ (shift=10)}$   $S'_{e\ (shift=30)}$   $S'_{e\ (shift=50)}$

**Steganalysis on deep Steg/hiding.** Like Baluja[1], we confirmed that UDH is robust to StegExpose steganalysis[2] but not to steganalysis DNNs. Baluja[1] and HiDDeN[25] showed a trade-off between capacity, secrecy (steganalysis), and/or robustness. This trade-off challenges DDH/UDH to hide a full image while deceiving steganalysis. The reviewer has the concern that deep Steg fails for steganalysis thus should not be called Steg, leading to the claim that "the core idea of the paper is incorrect". However, *Deep Steg* was widely used in prior works [1,24,25,26,29] and for consistency, we adopted the same term. However, "deep hiding" can be used to avoid confusion. Regardless of chosen terms, the success of deep Steg/hiding[1] is non-trivial, which inspires follow-up works, such as deep watermarking[29] and LFM[25], where steganalysis is **not** a major concern, instead robustness to corruptions[29] and light effect[25] are the major concerns. Despite their impressive performance, the mechanisms of deep hiding remain mostly unexplored and UDH, disentangling $C$ and $S$, is the first work to provide a frequency explanation towards a better understanding.

**UDH Description.** Lines90-95 describe the UDH training along with mentioned lines97-109. Fig. 1 & 2 shows the core difference between UDH & DDH. More granular details are given in Sec. 2 of the supplementary along with the code. Performance-wise, UDH is comparable to DDH for Steg (arguably slightly better and worse than Baluja[1] and R4's [9], respectively).

**Novelty.** We proposed the first universal hiding meta-architecture UDH that (1) first explains the success of deep Steg (2) first achieves (DNN-based) universal watermarking and outperforms HiDDeN (3) first achieves hiding an image for LFM and outperforms [25] without collecting additional dataset. Overall, our proposed UDH is simple yet versatile. **Sec3.2.** Different from R4's [9], our work shows the possibility of hiding M in N images; R4's [9] did not explore different recipients getting different secret messages.

## Footnotes

[1]We provide the link for the "perspective warp" function: https://pastebin.com/FsAC8EHu.


[Meta-Review · NeurIPS 2020]

Although the term steganography might be misleading for this paper, most reviewers agree that there is enough contribution for publication.